# Fast Amortized Fitting of Scientific Signals Across Time and Ensembles via Transferable Neural Fields

Sophia Zorek[1]    Kushal Vyas[1]    Yuhao Liu[1]    David Lenz[2]
Tom Peterka[2]    Guha Balakrishnan[1]

[1]Rice University
[2]Argonne National Laboratory

## Abstract

*Neural fields, also known as implicit neural representations (INRs), offer a powerful framework for modeling continuous geometry, but their effectiveness in high-dimensional scientific settings is limited by slow convergence and scaling challenges. In this study, we extend INR models to handle spatiotemporal and multivariate signals and show how INR features can be transferred across scientific signals to enable efficient and scalable representation across time and ensemble runs in an amortized fashion. Across controlled transformation regimes (e.g., geometric transformations and localized perturbations of synthetic fields) and high-fidelity scientific domains—including turbulent flows, fluid–material impact dynamics, and astrophysical systems—we show that transferable features improve not only signal fidelity but also the accuracy of derived geometric and physical quantities, including density gradients and vorticity. In particular, transferable features reduce iterations to reach target reconstruction quality by up to an order of magnitude, increase early-stage reconstruction quality by multiple dB (with gains exceeding 10 dB in some cases), and consistently improve gradient-based physical accuracy.*

## 1. Introduction and Background

Scientific domains frequently produce high-dimensional data spanning space, time, and simulation ensembles. Stemming from fields like climate modeling, computational fluid dynamics, and cosmology, these signals exhibit complex phenomena, irregular geometries, and heterogeneous transformations. Traditional grid-based structures (e.g., voxel lattices) cannot scale to meet the representation and visualization demands of these high-resolution observations. Consequently, there is significant interest in developing AI-driven frameworks specifically designed to tackle the prac-

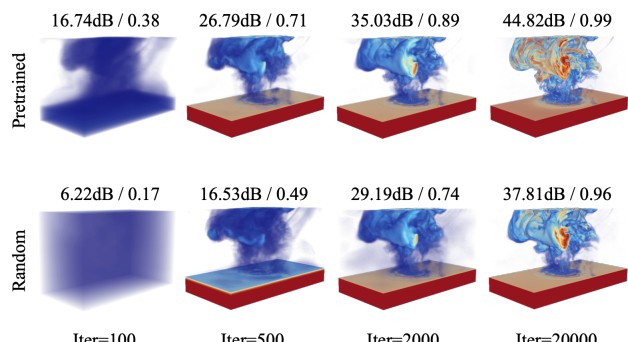

Figure 1. **Early reconstruction advantage from transferable features** (PSNR/SSIM). Water volume fraction from **Deep Water Asteroid Impact** is visualized with pretrained initialization (top) and random initialization (bottom). PSNR is averaged across all output fields. Pretrained models recover higher frequency detail of the water structure significantly earlier in training, yielding higher PSNR **(+10dB gain)** at early iterations.

tical challenges of modern scientific data [1].

Neural fields, often referred to as implicit neural representations (INRs), are a powerful family of continuous, learned function approximators for signal data. An INR maps a coordinate within a domain to one or more output values using a complex, learnable nonlinear function. To handle exceptionally large signals, the underlying multilayer perceptron (MLP) is often coupled with an efficient lookup data structure, such as a hash table [11]. INRs are highly attractive data structures because they: (1) enable arbitrary signal super-resolution agnostic to the sampling rate, (2) readily encode high-frequency details [14, 15, 24], (3) provide powerful priors for solving inverse problems [14, 15], and (4) can compress signals [5, 10]. For these reasons, over the past five years, INRs have been of significant research interest to a variety of research fields, from computer vision to scientific data representation and visualization [17, 18, 22, 23]

One key challenge with fitting INRs to large new signals is the time and computational costs required to train their parameters from scratch. If initialized with random parameters, an INR must typically be optimized for several thousands of iterations before reaching a level of reconstruction fidelity at which the dominant structure of the signal is reliably captured. Several recent studies instead show that data-driven approaches to parameter initialization—relying on prior training signals via meta-learning [19], hyper-networks [4, 15], and transfer learning [16, 21] —significantly outperform standard random sampling strategies. By learning shared structures across a dataset, these methods essentially *amortize* the expensive fitting cost over multiple signals. A particularly attractive transfer learning method in this vein is *Strainer* [20], which divides an INR into a shared encoder and $M$ individual decoder heads, training the network jointly on a set of $M$ real signals. At test time, the pretrained encoder is combined with a randomly initialized decoder to fit novel signals with remarkable efficiency, avoiding the need to relearn global features from scratch.

While demonstrated successfully across images from typical imaging datasets (e.g., faces and 2D MRI scans), INR transfer learning strategies such as *Strainer*, remain relatively unexplored for scientific signals. However, they have the potential to be particularly effective for this type of data, which often exhibit strong correlations across time and across simulation instances. Though meta-learning and hypernetwork-based approaches similarly aim to transfer knowledge across tasks or signals, they can be computationally expensive and difficult to scale to high-dimensional data. Motivated by this potential, it is unclear whether these light-weight strategies provide an effective mechanism to reduce optimization costs across time and simulation runs of scientific signals. Furthermore, the original *Strainer* framework focused solely on MLP architectures that encode global structures via smooth activation functions (e.g., SIREN). Large multi-scale scientific signals, however, often benefit from hybrid INR architectures with scalable hash grids or $K$-planes, leaving it an open question whether light-weight feature transfer translates to these models. Finally, while evaluation of INRs on natural images has largely focused on zero-order visual reconstruction metrics (e.g., PSNR), scientific applications demand a higher standard of physical fidelity, frequently requiring derived quantities, such as spatiotemporal gradients, to also be accurate. Therefore, there is also a critical understanding gap in how INR architectural choices and feature reuse strategies influence signal representation beyond basic reconstruction accuracy.

In this study, we explore the viability of transferable features for INRs to enable *fast, amortized fitting* of high-dimensional scientific data. We evaluate this along two primary axes. First, we measure reconstruction performance in terms of signal fidelity and convergence speed, quantifying both the final reconstruction accuracy (e.g., PSNR) and the number of iterations required to reach a target level of fidelity. Second, we assess the extent to which learned representations preserve physically meaningful structure by evaluating the accuracy of derived quantities, such as spatiotemporal gradients and other first-order features. We conduct controlled experiments on synthetic signals with diverse transformation regimes, including rotations, warping, and local perturbations. We then extend our analysis to physics-based datasets spanning 4D and 5D domains, including time-varying and ensemble-based simulations. Across these settings, we show that transferable features consistently accelerate convergence—reducing the number of iterations required to reach target reconstruction quality—while also improving the fidelity of derived physical quantities. These results demonstrate that feature reuse provides a scalable and structure-aware approach to modeling high-dimensional scientific signals.

## 2. Methods

### 2.1. Neural Representations for Scientific Signals

We model scientific data as continuous functions over high-dimensional coordinate spaces. Let $\mathbf{x} \in \mathbb{R}^d$ denote input coordinates, which may include spatial, temporal, and simulation dimensions (e.g., $d = 4$ or 5), and let $\mathbf{y} \in \mathbb{R}^C$ denote a set of physical fields such as density, velocity, or pressure. Implicit neural representations (INRs) parameterize this mapping as a neural function $f_\theta : \mathbb{R}^d \to \mathbb{R}^C$.

We consider three classes of INR models commonly used for scientific data, each introducing different trade-offs between smoothness, locality, and computational efficiency.

**Smooth function-based models** represent signals using deep multilayer perceptrons with continuous nonlinearities, such as sinusoidal activations. They produce globally smooth functions, and are well-suited for modeling continuous physical quantities and their derivatives [12, 15]. However, they are computationally expensive to optimize and can struggle to scale to high-dimensional domains.

**Spatial encoding-based models** map input coordinates to multiresolution feature embeddings (e.g., hash grids), followed by a shallow network. This enables efficient, high-fidelity fitting of large-scale data, but the use of local grid interpolation (e.g., trilinear) yields representations that are continuous but only piecewise smooth, with limited higher-order regularity [11].

**Factorized representations**, like K-Planes [6], decompose signals into lower-dimensional components, such as learned planar features over coordinate pairs. This significantly reduces memory and computation while preserving key structure, but introduces additional assumptions about separability across dimensions. These representations pro-

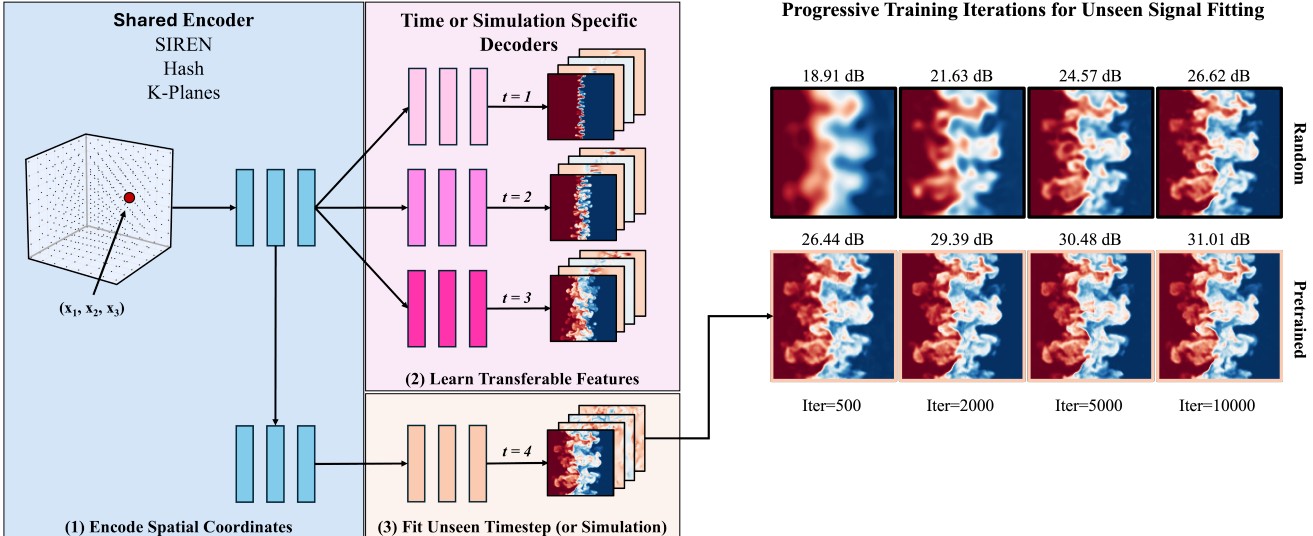

Figure 2. **Transferable neural fields for efficient fitting of 4D/5D signals.** A shared encoder learns reusable multiscale structure across signals, while lightweight time- or simulation-specific decoders enable efficient adaptation to new instances. This decomposition allows features learned from prior signals to be reused, providing a strong initialization for fitting unseen data **(left)**. Reconstruction progression (PSNR) for an unseen simulation from **Rayleigh–Taylor instability** using SIREN, with pretrained initialization (bottom) and random initialization (top). Density from a late-timestep slice is visualized. Pretrained models recover mixing structures significantly earlier, achieving higher PSNR at early iterations and resolving sharper interface dynamics **(right)**.

vide complementary inductive biases. Smooth models favor global structure and differentiability, while encoding-based and factorized models emphasize efficiency and local adaptability. This distinction becomes critical when modeling high-dimensional scientific data, where both scalability and physical fidelity are required.

## 2.2. Learning Transferable Features Across Signals

Scientific signals evolving over time or across simulation instances arise from shared underlying processes and can exhibit common multiscale structure. Recognizing these shared underlying dynamics provides a natural motivation for feature reuse. Given a collection of related signals $\{S_j\}_{j=1}^M$ across time or simulations, we aim to learn a shared representation that captures common structure across the dataset. We decompose some neural representation model into a shared encoder $f_\psi$ (with shared parameters $\psi$) and signal-specific decoders $g_{\phi_j}$ (with parameters $\phi_j$ for each signal $j$). The encoder maps coordinates to latent features, and each decoder maps these features to the output of the $j$-th signal. This separation allows the model to capture common structure in the encoder while retaining flexibility through the decoders. We train the model jointly across all signals by minimizing the reconstruction error:

$$(\psi^*, \{\phi_j^*\}) = \arg\min_{\psi,\{\phi_j\}} \sum_{j=1}^M \sum_i \mathcal{L}\big(g_{\phi_j}\big(f_\psi(\mathbf{x}_i)\big),\, S_j(\mathbf{x}_i)\big).$$
$$(1)$$

where $\mathbf{x}_i$ may include only spatial coordinates, with signals indexed by $j$ corresponding to different timesteps or simulations, or may explicitly include time as an input dimension, yielding a continuous spatiotemporal representation. Our framework supports both settings.

As illustrated in (Fig. 2), the training procedure encourages the encoder to learn features that are invariant across signals while allowing the decoders to capture time or simulation specific variations.

## 2.3. Efficient Fitting with Transferable Features

Given a related unseen signal $S_{\text{new}}$, we initialize the model using the learned shared encoder parameters $\psi$ and a randomly initialized decoder with parameters $\phi_{\text{new}}$. The model is optimized to fit the new signal:

$$(\psi^*, \phi_{\text{new}}^*) = \arg\min_{\psi,\phi_{\text{new}}} \sum_i \mathcal{L}(g_{\phi_{\text{new}}}(f_\psi(\mathbf{x}_i)),\, S_{\text{new}}(\mathbf{x}_i)).$$
$$(2)$$

The pretrained encoder provides a strong initialization, leading to faster fitting, improved reconstruction quality, and more stable optimization in high-dimensional settings. Importantly, this formulation is model-agnostic and extends across different INR architectures: in smooth models, transfer corresponds to reusing global functional structure; in encoding-based models, it corresponds to reusing spatial feature embeddings; and in factorized models, it corre-

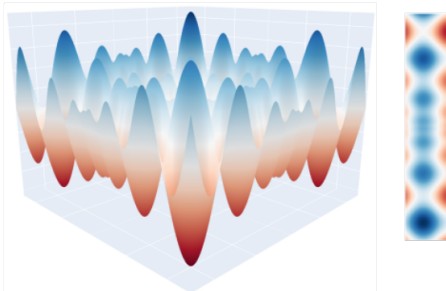
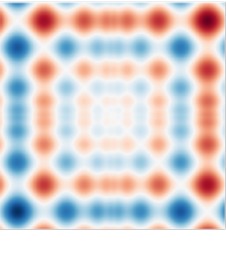

Figure 3. **2D Schwefel function** landscape (left) and top-down view (right). The function provides a non-convex landscape with rich local structure, making it well-suited for evaluating gradient fidelity under controlled transformations.

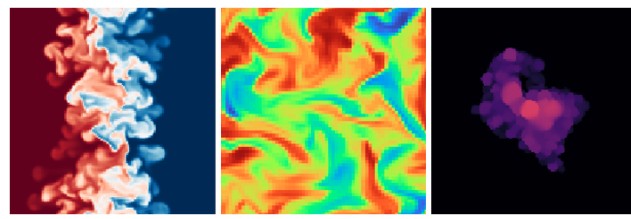

Figure 4. Representative slices from three datasets in *The Well* benchmark. **Rayleigh–Taylor instability** (left), **magnetohydrodynamics** (middle), and **supernova explosion** (right), illustrate diverse nonlinear and multi-scale spatial structures, providing a challenging setting for evaluating transferability across distinct physical regimes.

sponds to reusing low-dimensional projections. In high-dimensional scientific data, where signals evolve over time or across simulations, this enables scalable modeling by amortizing learning across related signals rather than fitting each instance independently.

## 3. Data

We consider three classes of data—controlled synthetic signals, physics-based benchmark simulations, and large-scale multiphysics systems—spanning increasing levels of complexity and realism, enabling systematic analysis of transfer under known transformations, nonlinear multi-physics regimes, and validation on heterogeneous physical systems.

### 3.1. Time-Evolving Toy Signal

We construct a controlled 3D dataset (2D space + time) based on the **Schwefel function** to systematically probe how neural representations capture shared structure across related signals. The base field is defined on a $500 \times 500$ grid:

$$f(x_1, x_2) = \sum_{d \in \{x_1, x_2\}} \left( 418.9829 - d \sin(\sqrt{|d|}) \right) \quad (3)$$

The function exhibits non-convex, multi-frequency structure (Fig. 3), making it well-suited for evaluating representation capacity. We generate a sequence of $T = 6$ timesteps by applying progressively stronger transformations to the coordinates or field values. We group the transformations into two categories: geometric (**Rotation** and **Warp**) and local (**Gaussian** and **Wave**).

Geometric transforms apply spatial transforms to a signal, essentially preserving structure but shifting locations of features (Fig. 8 in Supplementary). We define rotation and warp by:

$$\begin{pmatrix} x_1 \\ x_2 \end{pmatrix} = \begin{pmatrix} \cos\theta(t) & -\sin\theta(t) \\ \sin\theta(t) & \cos\theta(t) \end{pmatrix} \begin{pmatrix} x_1 \\ x_2 \end{pmatrix}, \quad (4)$$

$$\begin{pmatrix} x_1' \\ x_2' \end{pmatrix} = \begin{pmatrix} x_1 + \alpha(t)\sin(0.01\,x_2) \\ x_2 + \alpha(t)\sin(0.01\,x_1) \end{pmatrix}, \quad (5)$$

where $\theta(t)$ and $\alpha(t)$ vary smoothly over time.

Local transforms introduce spatially localized perturbations that change structure morphologies (Fig. 9 in Supplementary). We define a time-dependent local transform operator $P_t(x_1, x_2)$, and construct the transformed field as:

$$f_t(x_1, x_2) = f(x_1, x_2) + P_t(x_1, x_2).$$

We instantiate $P_t$ as either a floating Gaussian window:

$$P_t^{\text{Gauss}}(x_1, x_2) = A \exp\left( -\frac{(x_1 - c_1(t))^2 + (x_2 - c_2)^2}{\sigma} \right), \quad (6)$$

where $(c_1(t), c_2)$ controls the perturbation location over time and $\sigma$ controls the spatial window, or a wave:

$$P_t^{\text{Wave}}(x_1, x_2) = A\,w(x_1, x_2) \sin\big(0.05x_1 + 0.05x_2 + \psi(t)\big), \quad (7)$$

where $w(x_1, x_2)$ is a Gaussian window and $\psi(t)$ introduces a time-dependent phase shift.

### 3.2. Basic Physics Simulations (The Well)

We evaluate transferability on high-dimensional physical simulations from *The Well* [13] (Fig. 4), a collection of time-evolving scientific datasets generated from numerical solvers of coupled partial differential equations (PDEs). These datasets capture complex, nonlinear spatiotemporal dynamics across a range of physical systems, providing a realistic testbed where shared structure exists across simulations but is not trivially aligned.

**Rayleigh–Taylor Instability** [3] consists of a heavy fluid layered above a lighter fluid under gravity, leading to an unstable interface that evolves into turbulent mixing. The system exhibits rapid growth of small perturbations into

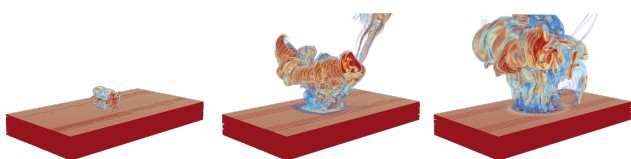

Figure 5. **Deep Water Asteroid Impact** simulation showing the time evolution of water volume fraction across timesteps (left to right). These high-resolution signals span complex, strongly interacting physical systems, enabling evaluation of transferability in more realistic, large-scale scientific settings.

complex, turbulent structures over time. The data is represented as a 5D tensor (simulation, time, $x_1$, $x_2$, $x_3$) with density and velocity fields, and has a spatial resolution of $128^3$.

**Magnetohydrodynamics (MHD)** [2] describes the interaction between fluid flow and magnetic fields through a coupled system of fluid and electromagnetic equations in the compressible, isothermal limit. The resulting dynamics involve multi-scale interactions between velocity, density, and magnetic field components. The data is similarly 5D, with multiple coupled output fields (e.g., $\rho$, $\mathbf{v}$, $\mathbf{B}$). The simulation has a spatial resolution of $64^3$.

**Supernova Explosion** [8] [9] simulations model the evolution of a supernova blastwave generated by injecting thermal energy into a dense gas cloud. The system evolves under compressible gas dynamics with additional physical processes such as cooling, leading to rapid expansion and shock formation. The resulting dynamics are highly nonlinear and span multiple scales over time. The data is represented as a 5D spatiotemporal tensor with output fields including density, pressure, temperature, and velocity, and has a spatial resolution of $128^3$.

### 3.3. Complex Physics Simulation (Asteroid)

We further evaluate on a large-scale multiphysics asteroid impact simulation, assessing transfer in complex, strongly interacting physical systems. **Deep Water Asteroid Impact** (Fig. 5) is a high-resolution multiphysics simulation of an asteroid impact into the ocean, modeled using a three-dimensional hydrocode with adaptive mesh refinement and tabulated equations of state. The simulation captures coupled interactions between the asteroid, ocean, and atmosphere, including shock formation, vaporization, and fluid–structure dynamics [7]. We focus on a 4D spatiotemporal volume $(t, x_1, x_2, x_3)$ defined on a $300^3$ spatial grid evolving over time. The dataset contains multiple physical fields, including density $(\rho)$, pressure $(P)$, temperature $(T)$, speed of sound (snd), volume fractions (water and asteroid material), and velocity components $(v_x, v_y, v_z)$. These variables jointly describe the coupled thermodynamic, material, and flow dynamics induced by the impact.

## 4. Experiments

We evaluate transferable features across multiple implicit neural representation (INR) architectures, including smooth function-based models (SIREN), spatial encoding-based models (hash grids), and factorized representations (K-Planes). For each dataset, we configure models with comparable parameter counts to ensure consistent compression ratios across architectures, defined relative to the total number of scalar values in the underlying spatiotemporal and multivariate data.

We train all models using Adam optimizer with dataset-specific learning rates selected for stable convergence. Hyperparameters are tuned per model and dataset to establish strong baselines, and then held fixed when comparing random initialization and pretrained transfer. Additional information regarding training details and hyperparameter tuning can be found in Section 7 (Supplementary). We train models to convergence, determined by saturation of reconstruction metrics. When fitting to an unseen signal, we initialize the shared encoder with pretrained weights and randomly initialize the decoder. We then optimize the model under the same training configuration as its fully random initialized counterpart (Figure 2).

We normalize input coordinates to fixed ranges ($[-1, 1]$ for SIREN and $[0, 1]$ for the others), and scale output fields per variable to $[-1, 1]$ using global min-max normalization. We evaluate performance using both reconstruction and convergence metrics. We measure reconstruction quality using PSNR and SSIM, while assessing convergence with iterations to reach a target quality threshold. We further evaluate the fidelity of derived quantities, including gradient-based and physically relevant metrics, to assess whether transferable features preserve meaningful structure beyond zero-order pointwise accuracy.

### 4.1. Controlled Spatiotemporal Experiments

We first study transferable features in a controlled spatiotemporal setting, where we systematically vary the underlying signal through known transformations. We consider a sequence of time-evolving signals and evaluate transfer by fitting the final timestep ($t = 5$) under different initialization strategies. Specifically, we compare (i) random initialization, (ii) pretraining on a single timestep ($t = 0$), and (iii) joint pretraining across multiple timesteps ($t = 0$–4), where pretrained encoder weights are used to initialize the model before fitting the target signal. All models are trained under a fixed compression ratio (2x), ensuring consistent capacity across architectures. Derivatives are calculated analytically for ground truth data and with autograd for each neural representation. Across transformations, we show how pretrained initializations consistently improve both convergence speed (Figure 6 and Tab. 2) and final gradient fidelity (Table 1) compared to random initial-

Table 1. Final gradient error (**RMSE, iter = 4k**) when fitting $t{=}5$ for transformations of the time evolved **Schwefel function**. Pretraining improves first-order gradient fidelity across models, with joint pretraining ($t{=}0$–4) often achieving the lowest error. SIREN produces substantially more accurate derivatives overall, reflecting its smooth functional representation compared to piecewise-linear alternatives.

| Model | Init | Final Gradient Error (RMSE) $\downarrow$ | | | |
| | | **Warp** | **Gaussian** | **Rotation** | **Wave** |
| --- | --- | --- | --- | --- | --- |
| **SIREN** | random | 0.23 | 0.21 | 0.23 | 0.22 |
| | $t{=}0$ | 0.12 | 0.13 | 0.17 | 0.14 |
| | $t{=}0$–4 | **0.10** | **0.08** | **0.11** | **0.09** |
| **Hash** | random | 0.30 | 0.28 | **0.35** | 0.29 |
| | $t{=}0$ | 0.32 | 0.27 | 0.49 | 0.27 |
| | $t{=}0$–4 | **0.29** | **0.26** | 0.42 | **0.25** |
| **K-Planes** | random | 0.70 | 0.81 | 0.57 | 0.83 |
| | $t{=}0$ | 0.73 | 0.80 | 0.55 | 0.82 |
| | $t{=}0$–4 | **0.67** | **0.75** | **0.54** | **0.68** |

ization.

### 4.2. Transfer Across 5D Physical Simulations

We evaluate transfer across simulation instances using three datasets from *The Well* [13]: **Rayleigh–Taylor instability**, **magnetohydrodynamics (MHD)**, and **supernova explosion**. Each dataset consists of multiple simulations of the same physical system, each represented as a 4D spatiotemporal field with multiple output variables. For each dataset, we consider a set of six simulations across 10 uniformly spaced timesteps. We jointly pretrain a shared encoder on five simulations and then fit the remaining held-out simulation using the pretrained encoder weights, comparing against a randomly initialized baseline. This process is repeated across all combinations of held-out simulation, and results are averaged. All models for experiments involving *The Well* are trained under a fixed compression ratio ($53\times$) to ensure consistent capacity across datasets and architectures. Derivatives are computed using higher-order finite differences.

We visualise training progression results in Figure 2 and Figure 7, showing how pretraining enables substantially faster recovery of fine-scale structure. In Tables 3 to 5, we show that transferable features across simulation instances reduce the number of iterations required to reach reconstruction thresholds by up to an order of magnitude, while also improving physically relevant quantities, including density gradients and vorticity.

### 4.3. Temporal Transfer in High-Fidelity Multi-Physics Systems

We evaluate large-scale temporal transfer on the multiphysics **Deep Water Asteroid Impact** simulation. We pretrain a shared encoder on an initial timestep, and then fit subsequent timesteps using the pretrained initialization, comparing against randomly initialized baselines. We pretrain at $t{=}10$k and evaluate transfer at later timesteps spanning the simulation ($t{=}20$k, $30$k, $40$k, $50$k). All models are trained under the same configuration as in previous experiments, but with a compression factor of 10x, and results are averaged across output fields and reported across architectures. Specifically in Figure 1 and Table 6, we see how transferable features across time improves both early stage and final reconstruction quality. The results show that transferable features scale to high-fidelity multiphysics systems, remaining robust under substantial temporal evolution and complex structural changes.

## 5. Discussion

Our experiments demonstrate that transferable features provide a consistent and practical mechanism for accelerating INR fitting across a wide range of settings, from controlled transformations to high-dimensional scientific simulations. Across architectures, compression factors, and datasets, pretraining reduces the number of iterations required to reach reconstruction thresholds (Table 2), and can even improve final reconstruction quality (Table 6). These gains persist across both temporal transfer (4D) and simulation-level transfer (5D) (Tables 3 to 5), suggesting that shared structure across related scientific signals can be effectively amortized.

However, the magnitude and consistency of these benefits depend on the underlying representation. Smooth function-based models (SIREN) benefit the most from transfer, exhibiting both rapid convergence and consistently improved gradient fidelity. This aligns with their global, continuous parameterization, which allows pretrained features to generalize across transformations without requiring significant reconfiguration. In contrast, spatial encoding-based methods such as hash grids and factorized representations like K-Planes show more variable improvements, particularly under geometric transformations (e.g., rotation) and phase shifts as shown in Table 2, and Table 3. When initialized with pretrained parameters, these models appear less effective at leveraging shared structure when it occurs at different spatial locations. This occurs, for example, across simulation instances where initial condition variations induce geometric transformations or phase shifts. Because these spatial encoding-based representations are tied to local embeddings or decomposed coordinate planes, the effectiveness of reusing pretrained features is reduced when spatial alignment across signals differs.

Despite this, transfer remains beneficial overall, even for these architectures, especially in settings where structural correspondence is preserved (e.g., temporal evolution or simulation ensembles). Notably, encoding-based methods

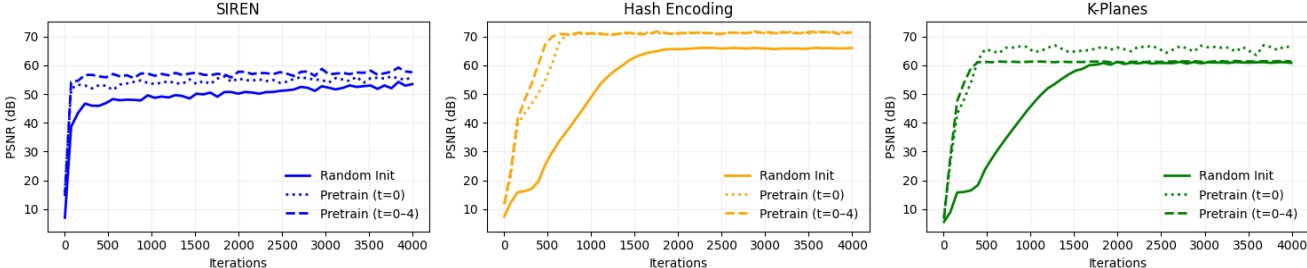

Figure 6. **PSNR (dB)** versus training iterations for fitting the final timestep ($t = 5$) under a local **Wave** transformation of the time evolved **Schwefel function.** Transferable features consistently accelerate convergence across all architectures, with joint pretraining across multiple timesteps ($t = 0$–4) often providing the strongest initialization.

Table 2. Iterations to reach reconstruction thresholds for **PSNR (dB)** and **SSIM** when fitting $t$=5 for three transformations of the time evolved **Schwefel function**. Pretraining consistently improves convergence across models and transformations, with largest gains from $t$=0–4. Benefits are reduced for Hash and K-Planes under geometric transformations (e.g., **Rotation**).

| Model | Init | Warp Iterations to Threshold ↓ | | Gaussian Iterations to Threshold ↓ | | Rotation Iterations to Threshold ↓ | |
|---|---|---|---|---|---|---|---|
| | | PSNR (50) | SSIM (0.9) | PSNR (50) | SSIM (0.9) | PSNR (50) | SSIM (0.9) |
| **SIREN** | random | 1836 | 73 | 1514 | 78 | 2236 | 74 |
| | $t$=0 | 157 | 61 | 76 | 70 | 636 | 58 |
| | $t$=0–4 | **67** | **36** | **43** | **38** | **156** | **43** |
| **Hash** | random | 873 | 398 | 1032 | 476 | **472** | 236 |
| | $t$=0 | 950 | 156 | 233 | 79 | 1038 | 315 |
| | $t$=0–4 | **397** | **73** | **214** | **70** | 644 | **157** |
| **K-Planes** | random | 956 | 476 | 956 | 476 | **798** | 467 |
| | $t$=0 | 876 | 156 | 156 | 76 | 2154 | 556 |
| | $t$=0–4 | **316** | **76** | **156** | **76** | 956 | **231** |

Table 3. **Rayleigh-Taylor instability** results from *The Well*. We report iterations required to reach reconstruction thresholds (**PSNR, SSIM**) and final derivative-based error (**RMSE, iters = 100k**) for density gradients ($\nabla \rho$) and vorticity ($\nabla \times \mathbf{v}$). Pretraining significantly accelerates convergence for SIREN and K-Planes and improves final physical fidelity. SIREN achieves the lowest gradient and vorticity error overall, reflecting its smooth functional representation.

| Model | Init | Iters to Threshold ↓ | | Final RMSE ↓ | |
|---|---|---|---|---|---|
| | | PSNR(30) | SSIM(0.9) | $\nabla \rho$ | $\nabla \times \mathbf{v}$ |
| **SIREN** | random | 2940 | 4039 | 1.02 | 0.73 |
| | pretrain | **208** | **587** | **0.86** | **0.62** |
| **Hash** | random | **418** | **1143** | **1.29** | **0.90** |
| | pretrain | 945 | 3193 | 1.35 | 0.94 |
| **K-Planes** | random | 22,155 | 38,325 | 2.72 | 1.69 |
| | pretrain | **1364** | **2170** | **2.49** | **1.59** |

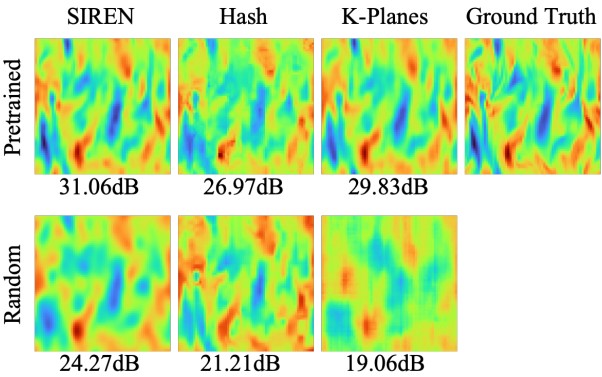

Figure 7. **MHD** reconstruction (**PSNR (dB)**) at 4k iterations on an unseen simulation, pretrained initialization (top) and random initialization (bottom). Density from a late timestep is visualized. Pretraining improves early-stage reconstruction, yielding sharper structures and improved density gradient fidelity across models.

can achieve strong zero-order reconstruction performance (e.g., PSNR), in some cases outperforming smooth models, but this often comes at the cost of reduced higher-order fidelity (Figure 6 and Tab. 1). As reflected in the gradient-based metrics of Table 3, piecewise-linear representations

Table 4. **Magnetohydrodynamics (MHD)** results from *The Well*. We report iterations required to reach reconstruction thresholds **(PSNR, SSIM)** and final derivative-based error **(RMSE, iters = 20k)** for density gradients ($\nabla\rho$) and vorticity ($\nabla \times \mathbf{v}$), reflecting coupled fluid and flow structure. Pretraining yields consistent speedups across all models, with particularly large gains for K-Planes, while SIREN achieves the lowest derivative error, reflecting improved preservation of flow structure and density variation.

| Model | Init | Iters to Threshold ↓ | | Final RMSE ↓ | |
|---|---|---|---|---|---|
| | | PSNR(25) | SSIM(.85) | $\nabla\rho$ | $\nabla \times \mathbf{v}$ |
| **SIREN** | random | 1498 | 1867 | 0.70 | 1.23 |
| | pretrain | **210** | **339** | **0.68** | **1.16** |
| **Hash** | random | 476 | 489 | 0.86 | 1.52 |
| | pretrain | **182** | **299** | **0.85** | **1.51** |
| **K-Planes** | random | 5054 | 1326 | 1.72 | 2.89 |
| | pretrain | **532** | **677** | **1.55** | **2.61** |

Table 5. **Supernova explosion** results from *The Well*. We report iterations required to reach reconstruction thresholds **(PSNR, SSIM)** and final physical error **(RMSE, iters = 100k)** for density gradients ($\nabla\rho$) and vorticity ($\nabla \times \mathbf{v}$). Pretraining substantially accelerates convergence for SIREN and K-Planes while improving derivative accuracy, whereas Hash shows inconsistent behavior, with minimal convergence gains and slight degradation in final error.

| Model | Init | Iters to Threshold ↓ | | Final RMSE ↓ | |
|---|---|---|---|---|---|
| | | PSNR(45) | SSIM(0.9) | $\nabla\rho$ | $\nabla \times \mathbf{v}$ |
| **SIREN** | random | 10,500 | 7889 | 0.15 | 0.12 |
| | pretrain | **1034** | **873** | **0.13** | **0.11** |
| **Hash** | random | **420** | **304** | **0.20** | **0.14** |
| | pretrain | 1043 | 779 | 0.21 | 0.15 |
| **K-Planes** | random | 61,530 | 48,764 | 0.49 | 0.30 |
| | pretrain | **8,112** | **6,237** | **0.48** | **0.41** |

produce less accurate derivatives, even when zero-order reconstruction quality is high.

Amortized training helps bridge this gap. By providing a strong initialization, transfer enables smooth models such as SIREN to achieve competitive reconstruction performance while retaining their advantage in derivative accuracy. This suggests that transferable features are particularly valuable not only for accelerating optimization, but also for improving the balance between zero-order fidelity and higher-order physical consistency. Overall, these results highlight an important trade-off between representation efficiency and smoothness, and show that transfer can also mitigate architecture-dependent limitations.

Table 6. **Deep Water Asteroid Impact** temporal transfer results. We report final **PSNR (dB, iter = 100k)** at increasing temporal distance $\Delta t$ from the pretrained state. A change of $\Delta t = 1$ corresponds to 1000 simulation timesteps. Pretraining consistently improves reconstruction quality, with gains decreasing as temporal distance increases.

| Model | Init | Final PSNR (dB) ↑ | | | |
|---|---|---|---|---|---|
| | | $\Delta t = 1$ | $\Delta t = 2$ | $\Delta t = 3$ | $\Delta t = 4$ |
| **SIREN** | random | 51.32 | 52.63 | 50.84 | **51.19** |
| | pretrain | **54.23** | **54.11** | **51.95** | 51.07 |
| **Hash** | random | 54.25 | 55.48 | 55.03 | 52.54 |
| | pretrain | **58.89** | **57.17** | **56.90** | **53.01** |
| **K-Planes** | random | 46.43 | 48.94 | 47.69 | 46.24 |
| | pretrain | **51.37** | **52.53** | **49.94** | **46.99** |

## 6. Conclusion

We present a systematic study of transferable features for implicit neural representations in high-dimensional scientific settings. Across controlled transformations, time-evolving signals, and multi-simulation datasets, we show that pretraining consistently accelerates convergence and improves final reconstruction quality, enabling efficient amortized fitting across both 4D and 5D domains (Tab. 3 and Tab. 6).

Our results further demonstrate that the effectiveness of transfer depends on the underlying representation. Smooth neural fields benefit most consistently, particularly for preserving derivative accuracy, while spatial encoding-based methods exhibit more variable gains when shared structure is not spatially aligned. Importantly, transferable features help bridge key trade-offs between fast convergence, high reconstruction fidelity, and physically meaningful derivative accuracy. Overall, this work highlights transferable neural fields as a promising direction for scalable scientific representation learning, particularly in regimes where signals exhibit shared structure across time or simulation ensembles. Future work may explore more robust mechanisms for handling spatial misalignment and extending transfer to broader classes of physical systems and real-world data.

## Acknowledgments

This material is based upon work supported by the U.S. Department of Energy, Office of Science, Office of Advanced Scientific Computing Research under contract number DE-AC02-06CH11357, Department of Energy Computational Science Graduate Fellowship under Award Number DE-SC0025528.

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

# Fast Amortized Fitting of Scientific Signals Across Time and Ensembles via Transferable Neural Fields

## Supplementary Material

## 7. Additional Training Details

### 7.1. Model Considerations

In addition to **SIREN** [15] and **K-Planes** [6], we use a modified **fhash** encoder [18] as our spatial encoding-based model. Fhash builds on multi-resolution hash encodings (e.g., tiny-cuda-nn [11]) and has been shown to be effective for representing spatiotemporal data. We adapt the fhash implementation to support 5D inputs for our experiments. We summarize the training settings for one dataset per category below.

### 7.2. Time-Evolving Toy Signal

We summarize training configurations for the Schwefel experiments described in Section 3.1 below (Table 7).

### 7.3. Rayleigh–Taylor Instability

We summarize training configurations for the Rayleigh–Taylor instability experiments described in Section 3.2 below (Table 8).

### 7.4. Deep Water Asteroid Impact

We summarize training configurations for **Deep Water Asteroid Impact** experiments described in Section 4.3 below (Table 9)

## 8. Dataset Visualization: Time-Evolving Toy Signal

We visualize the time-evolving Schwefel function under the transformations described in Section 3.1. Each transformation produces a sequence of signals with progressively increasing variation over time while preserving underlying structure. These controlled evolutions provide an interpretable setting for analyzing how representations capture and transfer structure across timesteps.

Table 7. Model and training settings for the time-evolving **Schwefel** dataset.

| Component | Value |
|---|---|
| Compression Ratio | $\sim 2\times$ |
| Training Iterations | 4,000 |
| Batch Size | 65,536 coordinates |
| Optimizer | Adam |
| Learning Rate | $1 \times 10^{-4}$ |
| Loss | $\ell_1$ |
| $\omega_0$ (for SIREN) | 10 |

Table 8. Shared training settings for the **Rayleigh–Taylor instability** dataset.

| Component | Value |
|---|---|
| Compression Ratio | $\sim 53\times$ |
| Training Iterations | 100,000 |
| Batch Size | 200,000 coordinates |
| Optimizer | Adam |
| Learning Rate | $1 \times 10^{-3}$ |
| Loss | $\ell_1$ |
| $\omega_0$ (for SIREN) | 30 |

Table 9. Shared training settings for the **Deep Water Asteroid Impact** dataset.

| Component | Value |
|---|---|
| Compression Ratio | $\sim 10\times$ |
| Training Iterations | 100,000 |
| Batch Size | 2,000,000 coordinates |
| Optimizer | Adam |
| Learning Rate | $1 \times 10^{-4}$ |
| Loss | $\ell_1$ |
| $\omega_0$ (for SIREN) | 30 |

# Rotation

# Warp

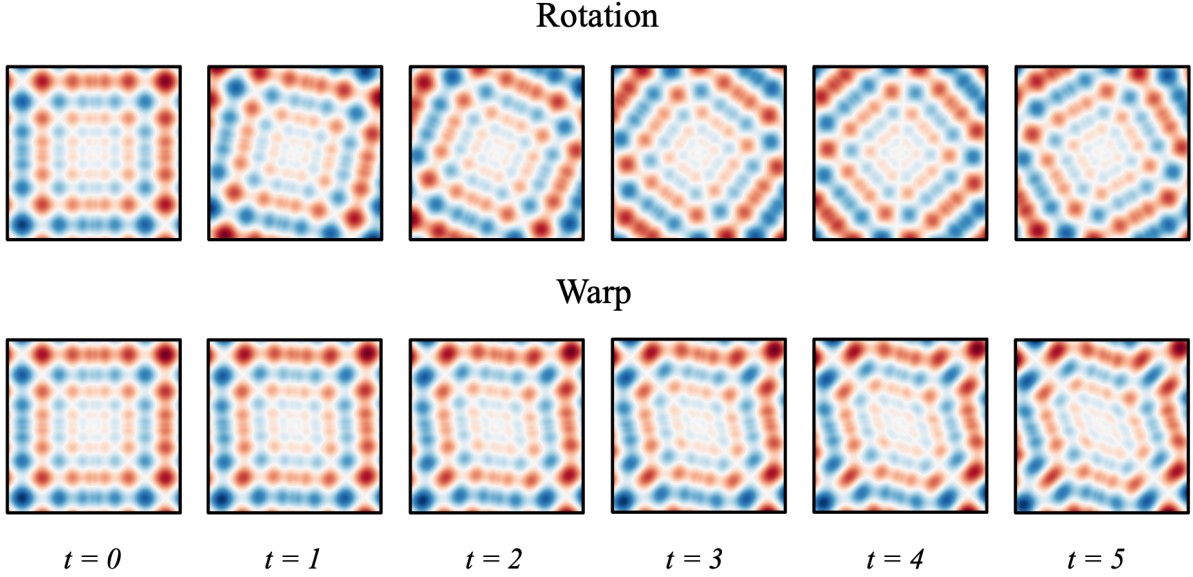

$t = 0$  $t = 1$  $t = 2$  $t = 3$  $t = 4$  $t = 5$

**Figure 8. Geometric transformations over six timesteps.** Each row corresponds to a different transformation: **Rotation** (top) (Eq. 4) and **Warp** (bottom) (Eq. 5). Time progresses from left to right. These transformations preserve underlying signal structure while altering spatial configuration, enabling evaluation of feature transfer under spatial misalignment.

# Gaussian

# Wave

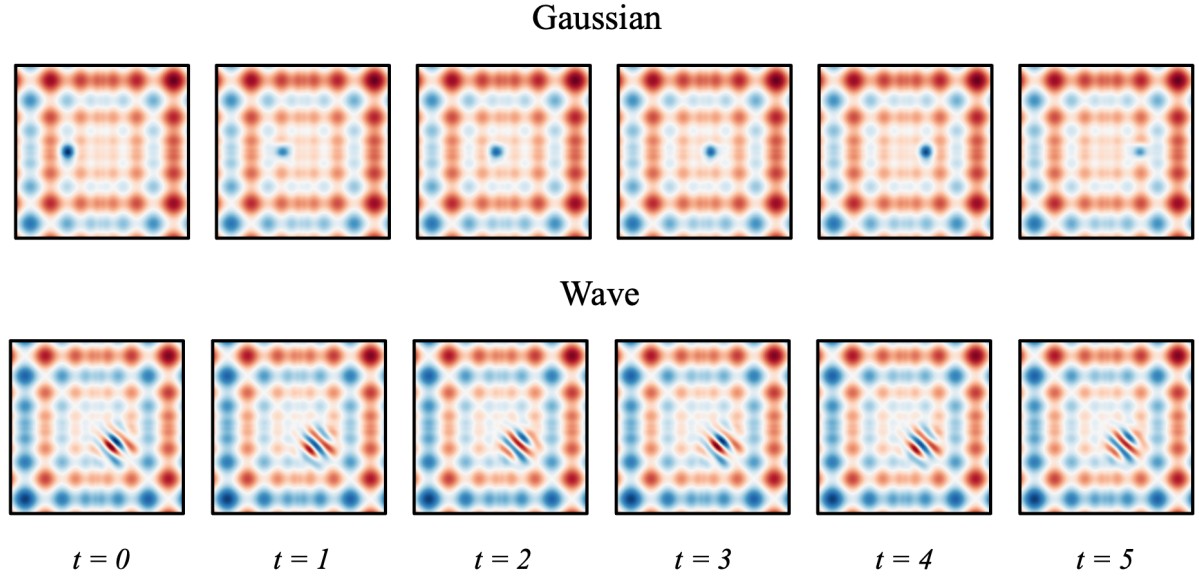

$t = 0$  $t = 1$  $t = 2$  $t = 3$  $t = 4$  $t = 5$

**Figure 9. Local transformations over six timesteps.** Each row corresponds to a different transformation: **Gaussian** (top) (Eq. 6) and **Wave** (bottom) (Eq. 7). Time progresses from left to right. These transformations emulate localized, time-evolving structures found in scientific systems, providing a controlled setting to assess transferability under known structural variations.

