# OpenReview forum: "Fast Amortized Fitting of Scientific Signals Across Time and Ensembles via Transferable Neural Fields"
_thecvf.com/CVPR/2026/Workshop/3D4S — CVPR 2026 Workshop 3D4S Poster_

### Official Review · Reviewer_nYYH · 2026-04-22
**Strong engineering value but lack of transferability**

**Rating:** 6
**Confidence:** 3

**Review:**

This paper investigates transferable neural fields for accelerating the fitting of high-dimensional scientific signals across time and simulation ensembles. By reusing pretrained encoder features, the method improves optimization efficiency for existing 4D/5D multivariate data and is supported by extensive experiments, demonstrating practical engineering value.

However, the paper has several limitations. First, the methodological novelty is moderate. Modeling time-varying multivariate physical fields is already well studied, and the core contribution mainly lies in applying feature reuse / warm-start transfer strategies to INR fitting rather than introducing a fundamentally new scientific modeling framework. Second, the experimental evidence mainly supports transferability among related signals within the same physical family, rather than broad generalization across heterogeneous physical domains. Third, most experiments are conducted on simulated benchmark datasets, with limited validation on real-world measured data.

---

### Official Review · Reviewer_CD4k · 2026-04-23
**Review of Transferable INR**

**Rating:** 6
**Confidence:** 3

**Review:**

### Summary
The paper studies transferable INR features for faster fitting of scientific signals. The empirical study is useful, especially because it evaluates derivative fidelity, but the method is incremental and the gains are not consistent across architectures.

### Strengths
- The paper studies related signals across time or ensembles should share reusable structure in INR. The evaluation is broader than simple PSNR, including gradient and physical-quantity fidelity.
- The controlled experiments are useful because they separate geometric transformations from local perturbations, making the behavior easier to interpret.

### Weaknesses
- The method itself is not very novel. A shared encoder with signal-specific decoders is a familiar transfer-learning formulation, and the paper’s contribution is mostly applying this idea to scientific INRs.
- If the claim is amortized fitting, it's best to compare meta-learning initialization, hypernetwork, learned optimizer, strainer, fine-tuning all weights vs. freezing encoder, pre-training full model vs. only encoder, etc. However, the paper mainly compares random initialization and pre-trained encoder. This baseline is somewhat low.

---

### Decision · Program_Chairs · 2026-04-28

Accept (Poster)